# Nursing Intervention to Prevent and Manage Delirium in Critically Ill Patients: A Scoping Review

**DOI:** 10.3390/healthcare12111134

**Published:** 2024-06-01

**Authors:** Filipa Fernandes, Mariana Santos, Ana Margarida Anacleto, Cátia Jerónimo, Óscar Ferreira, Cristina Lavareda Baixinho

**Affiliations:** 1Hospital de Vila Franca de Xira, 2600-009 Vila Franca de Xira, Portugal; ffernandes@campus.esel.pt (F.F.); ana.anacleto@hvfx.min-saude.pt (A.M.A.); catia.jeronimo@hvfx.min-saude.pt (C.J.); 2Hospital de Santa Maria, 1649-035 Lisboa, Portugal; marianasantos1@campus.esel.pt; 3Nursing Research, Innovation and Development Centre of Lisbon (CIDNUR), 1900-160 Lisbon, Portugal; oferreira@esel.pt; 4Center of Innovative Care and Health Technology (ciTechCare), 2414-016 Leiria, Portugal

**Keywords:** adult, critical care, critically ill, delirium, intensive care unit, nursing care

## Abstract

Delirium is an acute neuropsychiatric syndrome of multifactorial etiology with a high incidence in people admitted to intensive care units. In addition to reversible impairment of cognitive processes, it may be associated with changes in thinking and perception. If, in the past, it was considered an expected complication of severe disease, nowadays, delirium is associated with a poor short-term and long-term prognosis. Knowing that its prevention and early identification can reduce morbidity, mortality, and health costs, it is vital to investigate nursing interventions focused on delirium in critically ill patients. This study aimed to identify nursing interventions in the prevention and management of delirium in critically ill adults. The method used to answer the research question was a scoping review. The literature search was performed in the Medline (via PubMed), CINAHL (via EBSCOhost), Scopus, Web of Science, and JBI databases. The final sample included 15 articles. Several categories of non-pharmacological interventions were identified, addressing the modifiable risk factors that contribute to the development of delirium, and for which nurses have a privileged position in their minimization. No drug agent can, by itself, prevent or treat delirium. However, psychoactive drugs are justified to control hyperactive behaviors through cautious use. Early diagnosis, prevention, or treatment can reduce symptoms and improve the individual’s quality of life. Therefore, nursing professionals must ensure harmonious coordination between non-pharmacological and pharmacological strategies.

## 1. Introduction

Delirium is one of the most common mental illnesses in people hospitalized for an acute disease. According to figures from the National Institute for Health and Care Excellence [1], its prevalence is 30% in medical wards and 50% in surgical wards. However, it is in intensive care units (ICUs) that the prevalence and incidence of delirium are higher [2]. Studies report that up to 80% of critically ill people hospitalized in ICUs develop delirium at some point during their stay [3], and this condition is more prevalent in people receiving mechanical ventilation (60–80%) than in those who are spontaneously ventilated (20–50%) [4]. However, this remains an underdiagnosed condition, as its symptoms are incorrectly associated with other medical conditions [5].

There is a high number of stressors for the development of delirium, and its etiology is multifactorial [1]. It is estimated that this population is subject to 10 or more simultaneous risk factors [6]. Non-modifiable risk factors with strong evidence of an association with delirium are increasing age, pre-existing dementia, previous history of coma, emergency surgery, and trauma [7]. Precipitating risk factors are related to the acute pathology, its severity, and the environment, and are therefore modifiable and are the main target of nursing intervention. Iatrogenic precipitating risk factors include metabolic alterations, hydroelectrolytic imbalances, uncontrolled pain, and the administration of sedative drugs. In addition, the presence of invasive devices, sleep deprivation, immobilization, sensory hyperstimulation, noise, artificial lighting, temporal disorientation, visual and/or hearing impairment, and isolation from family are possible environmental precipitating factors [6].

Delirium is associated with a number of complications, such as prolonged hospital stays, increased risk of self-extubation and catheter removal, increased morbidity and mortality, and higher healthcare costs [8]. Given these consequences, and similar to other clinical conditions, prevention is better than treatment, since early intervention can reduce not only the incidence of delirium, but also its severity, the duration of symptoms, and the development of the aforementioned complications. 

Although the management of delirium requires coordinated multidisciplinary action, nurses play a fundamental role in the assessment, prevention, identification, and early resolution of delirium through non-pharmacological and pharmacological nursing interventions [2,5,9].

The aim of this scoping review is to map non-pharmacological and pharmacological nursing interventions aimed at preventing and managing delirium in adult/elderly people admitted to ICUs.

## 2. Materials and Methods

### 2.1. Study Design

An initial exploratory search of databases and protocol registration platforms for literature reviews revealed that there are several systematic reviews on the incidence of delirium in ICUs, as well as on risk factors and the association between pharmacological therapy and delirium prevention. However, no reviews or protocols were identified on nursing interventions in the prevention and management of delirium that include pharmacological and non-pharmacological measures.

The scoping review (SR) method was chosen because it allows for the identification of the evidence types available on the subject, examination of study types, and detection and analysis of knowledge gaps [10]. The study protocol followed the JBI recommendations [10,11].

The study protocol was registered on the Open Science Framework (OSF) platform under the registration: doi:10.17605/OSF.IO/UW2SC.

### 2.2. Research Question and Eligibility Criteria

The research question was structured according to the PCC (Population, Concept, and Context) mnemonic (Peters et al., 2020 [10]): “What nursing interventions are effective in preventing and managing delirium in critically ill people admitted to an intensive care unit?” and determined the eligibility criteria used in the process of searching for, selecting, and analyzing scientific evidence (Table 1). 

Articles were restricted to full text, open access, available in English, Portuguese, or Spanish. The time frame set was 2018–2023, due to the timeliness of the information and in order to structure recommendations for clinical practice based on the most recent evidence available on the subject.

### 2.3. Study Identification

The literature was searched in the following databases: Medline (via PubMed) (Table 2), CINAHL (via EBSCOhost), Scopus, Web of Science, and JBI. The search strategy used combined natural language with indexed language, and it should be emphasized that the indexed terms were duly adapted to the specificities of the databases. Each bibliographic reference obtained was then exported to the Rayyan^®^ application. 

In the second phase, a new literature search was performed through the bibliographic references of our research. Additionally, a search in the gray literature was conducted [10]. 

The entire process of finding and selecting the studies is explained in the results section, according to the PRISMA 2020 flow diagram for new systematic reviews, which included searches in databases, records, and other sources [12]. 

### 2.4. Data Processing and Analysis

Two researchers independently (F.F., M.S.) screened the articles based on the inclusion and exclusion criteria, and a third researcher (C.L.B.) adjudicated any disagreements. 

A Microsoft Excel^®^ (version 16.85) table was created to extract the content from the final bibliographic sample. It contained the following information: article title, author name(s), publication year, article type, objectives, methods, and main results/conclusions. 

After the extraction of results, a thematic synthesis was carried out to organize the interventions according to their nature.

## 3. Results

Of the 170 articles identified, 33 were duplicate references. A total of 137 articles were analyzed according to their title and abstract, and 104 were excluded. A total of 33 articles were read in full, 20 of which were rejected. The gray literature search identified 2 articles that met the eligibility criteria, and the final bibliographic sample consisted of 15 articles. The detailed process is presented in Figure 1. 

Table 3 shows the studies that make up the final bibliographic sample and synthesizes the main results that answered the research question [2,3,5,13,14,15,16,17,18,19,20,21,22,23,24]. The included articles have different methodologies and are from the USA [2,3,13,17,18,22], South Korea [14,23], Iran [16], Germany [15], Portugal [5], Canada [19], Poland [20], China [21], and India [24].

Content analysis of the articles included in the bibliographic sample enabled the identification of two categories of nursing interventions for the prevention and control of delirium: non-pharmacological and pharmacological interventions.

### 3.1. Non-Pharmacological Interventions

Delirium is a clinical condition associated with multiple risk factors, which are divided into predisposing and precipitating factors. The latter, which will be the focus of our initial discussion, can be divided into iatrogenic and environmental factors. 

Environmental risk factors are related to the presence of invasive devices, sleep deprivation, immobilization, hyperstimulation or sensory deficits, temporal disorientation, and isolation from the family. Meanwhile, iatrogenic risk factors depend on changes related to the medical diagnosis that led to the hospitalization, as well as the administration of drugs.

The aforementioned risk factors are the main target of nursing interventions in the prevention and management of delirium and, as such, we will present a set of results found in recent scientific evidence. Regarding non-pharmacological interventions, these are divided into several subcategories, formulated in accordance with the approach to the risk factors listed.

The first subcategory of non-pharmacological interventions identified concerns the promotion of sleep patterns. According to Bento and Sousa, promoting a routine sleep–wake cycle is an essential part of nursing intervention in the prevention of delirium. This requires modifying care routines [5], minimizing the number of nocturnal interruptions [18], and prioritizing and grouping tasks in order to promote better satisfaction of the need to sleep and rest. Other authors have presented a set of sleep-promoting interventions: the use of earplugs and sleep masks; minimizing noise and light [18]; and the use of light consistent with day–night circadian cycles [23].

Environmental control is related to the promotion of sleep patterns. At this level, the main objective is to provide a comfortable physical environment, minimizing certain stressors. To this end, some authors identify a reduction in sound stimuli, such as alarm sounds, and a reduction in exposure to artificial light as nursing interventions [5,23]. This avoids sensory hyperstimulation, an important risk factor for the development of delirium.

As part of the prevention and management of delirium, it is essential to constantly reorient the individual in the four aspects: time, space, person, and situation, taking into account the current date and time, using clocks and calendars as material resources, which should be present in the unit. [2,5,23]. 

In this process of reorientation, Chou, Pogach, and Rock [22] highlight the importance of including the individual’s name in speech, as well as the names of loved ones, pets, or other familiar words. The family is a fundamental element that nurses can and should use to meet this need and, consequently, to prevent the development of delirium [18]. 

Another subcategory is cognitive and memory stimulation. In this case, the involvement of family and friends is crucial, particularly through visits, viewing photographs, discussing family life, and/or recalling past events [5]. This stimulation can also be performed by listening to music and/or radio programs of interest to the person, or by participating in other activities such as solving puzzles or reading books [18]. 

Finally, sensory stimulation or reducing sensory deficits is another subcategory mentioned by several authors. It is important to provide the person in a critical situation with a hearing and/or visual impairment, where applicable, with glasses and/or hearing aids to overcome their impairment [5,18]. In the study by Liang et al. [21], the participants mentioned that adopting a sensory stimulation protocol (through the presence of photographs of family members, making phone calls, or listening to audio) could improve the clinical results of people in critical condition, in terms of the incidence and severity of delirium.

Early mobilization is another emerging subcategory. At this level, there are interventions to encourage early mobilization, such as sitting, walking, and performing rehabilitation exercises [18,23]. 

Promoting comfort is another important subcategory in the prevention of delirium. If, up to now, the categories identified refer to interventions that act on environmental risk factors, at this level, nurses will already be acting on iatrogenic risk factors. Thus, interventions will include pain and discomfort management [2], including the prevention of dehydration and constipation, assistance with self-care, and regular anxiety relief [23].

Following the non-pharmacological interventions discussed, it was possible to see that they are in line with what is recommended in the ABCDEF bundle, mentioned by several authors, which focuses essentially on the prevention of delirium. The meaning of each of the letters in the mnemonic system was: (A) assessment, prevention, and management of pain; (B) awakening the individual and training spontaneous breathing; (C) selection of sedation and analgesia; (D) assessment, prevention, and management of delirium; (E) early mobilization; and (F) family involvement (Bento and Sousa, 2021). The application of this bundle has been proven to be effective in improving survival, length of stay, duration of delirium, reduction in health costs, and readmission to ICU and/or other institutions after discharge [2,5,23].

Among the sample of articles included, there is one that addresses a future perspective in terms of interventions that could help prevent delirium, including the presence of advanced neuromonitoring; environmental changes, such as separation between equipment and patient, to allow for noise control and the presence of natural light; connection with family and friends; presence of guiding elements (clocks and calendars); use of artificial intelligence; recording of brain activity; and videomicroscopy, a new technology that can detect dynamic cellular changes and may be useful in monitoring delirium [20]. 

### 3.2. Pharmacological Interventions

In Portugal, the administration of pharmacological therapy is an interdependent nursing intervention. Despite being interdependent, and the aim of the study being to identify effective nursing interventions in the prevention and management of delirium, the literature review would lack scientific relevance if pharmacological strategies were not discussed. 

The above statement is corroborated by the Pain, Agitation/Sedation, Delirium (PADIS) guidelines [23], which address pharmacological and non-pharmacological strategies for the prevention and treatment of pain, agitation/sedation, and delirium in ICU patients.

Despite the lack of scientific evidence that pharmacological therapy prevents or reduces the duration of delirium, pharmacotherapy is being used to control this complication. In addition, the literature describes its widespread use in the treatment of hyperactive delirium, with psychomotor agitation and risk of physical injury [18]. 

#### 3.2.1. Pharmacological Prevention of Delirium in the ICU

The pathophysiological mechanism of delirium is not fully understood [5]. However, excessive dopaminergic activity has been suggested as a possible contributing factor. The hypothesis of an imbalance in the number of neurotransmitters and associated cholinergic deficiency has guided studies aimed at assessing the benefit of antipsychotic drugs in delirium. Although these drugs antagonize the dopamine pathways, none have shown a significant reduction in this complication in critically ill individuals [2].

One study noted that, in fixed prophylactic doses, the use of haloperidol is currently not recommended, as it may increase the total pharmacological dose and the risk of side effects, namely extrapyramidal and cardiac effects [15].

A study was conducted with the aim of evaluating the efficacy of low-dose quetiapine (12.5 to 25 mg) in preventing delirium in critically ill individuals. Both the study drug and the placebo were administered orally once every 24 h. The incidence rate of delirium in the first 10 days of ICU stay was 46.7% in the quetiapine group and 55.0% in the placebo group. Special attention should be paid to the positive results in the Confusion Assessment Method for Intensive Care Units (CAM-ICU), which was significantly lower in the quetiapine group than in the placebo group (14.4% vs. 37.5%), the duration of delirium, which was considerably shorter in the study group than in the control group, and the success rate of ventilator weaning, in which more subjects in the quetiapine group were successful (84.6% vs. 47.1%) [14].

#### 3.2.2. Pharmacological Management of Delirium in the ICU

No single pharmacological agent can treat delirium. Multiple authors have dedicated themselves to the individualized study of drugs in the treatment of delirium. However, it is known that this complication results from a complex pathophysiological process, with a multifactorial etiology and a dynamic interrelationship between preexisting vulnerability and iatrogenic factors. Researchers have therefore questioned whether it would be more effective to act in a concerted manner, through a bundle of pharmacological management [15]. 

Drug treatment is certainly useful in controlling hyperactive behavior in delusional individuals. However, the management of this clinical condition requires a skillful combination of non-pharmacological and pharmacological strategies [2]. Regardless of the subtype of delirium, there is evidence that haloperidol is the drug of choice for its treatment. Other agents commonly used in the management of delirium are benzodiazepines (36.0%), dexmedetomidine (21.3%), quetiapine (18.5%), and olanzapine (8.6%) [15].

Although in the clinic, haloperidol continues to be one of the first choices for the treatment of delirium, the evidence no longer recommend is use, as it is not associated with shorter durations of delirium, mechanical ventilation, or ICU stay, or reduced mortality. Currently, although its efficacy has not been exhaustively studied, it is suggested that haloperidol be used in the short term [2] in cases of delirium in which individuals experience forms of significant distress secondary to the symptomatology, such as agitation, anxiety, fear, hallucinations, or delusions, but not systematically in all critically ill individuals. 

There is a scientific consensus regarding the exacerbation of the risk of delirium with continuous infusion of benzodiazepine drugs, although it is accepted that the depth of sedation and the pharmacological dose prescribed may be more significant risk factors than the drug selected. Thus, some health professionals assume that benzodiazepines should be avoided in the treatment of delirium, while others report that they are useful in the treatment of individuals with ethanol habits, or in the case of delirium resulting from seizures [3]. 

Further research into combined pharmacological approaches is desirable in the future [22]. The literature reports that hyperactive delirium is less prevalent than the hypoactive subtype. However, the former not only carries a greater number of risks, but also requires increased nursing surveillance. Agitated individuals are often subjected to chemical and/or mechanical restraint methods, but these approaches carry a greater risk of complications. Some authors are, therefore, looking for alternatives to conventional medicine to help control agitation in delirious individuals. The use of herbal medicine is one of the examples explored. 

One study evaluated the efficacy and safety of valerian and lemongrass in addition to quetiapine in critically ill individuals diagnosed with delirium. The treatment lasted five days, during which 5 mL of Neurogol (oral solution of valerian and lemongrass) was administered concomitantly with the antipsychotic drug. The therapeutic group showed a significant improvement in agitation and a reduction in the length of stay in the ICU. It is believed that the addition of Neurogol to an antipsychotic drug regimen enhances its effect in controlling delirium, without unfavorable side effects [16]. 

Although the anxiolytic mechanisms of valerian and lemongrass are unclear, according to this study, Neurogol is considered an effective modality in the management of delusional individuals. However, more research is needed to investigate the role of herbal medicines [16]. 

## 4. Discussion

The bibliographic sample of this scoping review allows for an answer to the research question. Pharmacological and non-pharmacological interventions have been identified for the prevention and control of delirium in patients in intensive care. There is no consensus on which drugs can be used to prevent or treat delirium. Although drug therapy should not be administered routinely, it is useful for controlling hyperactive behavior. Atypical antipsychotics have gained increasing relevance, particularly in the population susceptible to extrapyramidal effects. However, haloperidol remains the most commonly used drug in the treatment of delirium [2,7,14,16,22], even though its use is not supported by the results of other studies [25]. 

According to PADIS (2018), antipsychotics, dexmedetomidine, statins, and ketamine are not recommended for the prevention of delirium. However, some recently published studies report the prophylactic potential of drugs such as quetiapine, dexmedetomidine, suvorexant, and exogenous melatonin [25]. 

Dexmedetomidine has an advantageous profile in the management of agitated individuals under difficult extubation or ventilator weaning. Benzodiazepines, although controversial, are thought to be suitable in individuals with ethanol habits or delirium underlying seizures. Herbal modalities have also shown some benefits [7,14,16,22,26].

Although nurses are not responsible for prescribing this type of therapy, they are in charge of administering it and identifying side effects early on, playing a leading role in the assessment, screening, and treatment of delirium [27]. 

This finding is corroborated by Von Rueden et al. [27] and Donovan et al. [28], who consider nurses’ interventions to be crucial in the assessment and treatment of delirium, because the continuous monitoring of patients in the ICU makes it possible to identify risk factors and associated symptoms at an early stage, while decision making in relation to communication with medical teams and patients makes it possible to intervene in an interdisciplinary manner at an early stage. The combination of pharmacological and non-pharmacological measures allows for the modification of certain risk factors and a reduction in the incidence of delirium [29].

When planning patient care, nurses should not just consider the administration of prescribed therapy. They must develop the skills to coordinate medication administration schedules, monitor vital parameters, and assess side effects with other procedures. These include non-pharmacological procedures for the prevention and management of delirium, such as controlling excessive light and noise so that adequate quality and quantity of sleep can be promoted [18,20,30]. The results of this scoping review reinforce the finding of Hoch et al. [30] that there is sufficient evidence that a multicomponent non-pharmacological approach can effectively prevent the onset of delirium and reduce symptom duration.

Regarding non-pharmacological measures, the results recommend pain assessment, prevention and management, spontaneous breathing training, early mobilization, and family involvement [2,5,23,29,31]. 

One study investigated the introduction of a multicomponent nursing intervention with cognitive stimulation and family support that was performed daily for about 15 min from admission to discharge from the ICU by two previously trained nurses. The results demonstrated the effectiveness of the intervention, shown by a low incidence of delirium in the critically ill patients who received the intervention, with an incidence rate of delirium in the control group of 20.1% and 33.1 per 1000 person-days (CI 95% 22.7 to 48.3) and in the intervention group of 0.6% and 0.64 per 1000 person-days (CI 95% 0.22 to 11.09) [31].

It is recommended that future studies explore the experience of these inpatients and their families in order to identify interventions to meet the needs and preferences of these patients [32]. Further research is needed about the knowledge and skills of professionals for the prevention, early detection, and treatment of delirium. Studies have observed that a lack of knowledge and experience in this area can lead to nurses not correctly detecting the risk of developing delirium and, consequently, not applying the necessary preventive measures. Thus, according to Liang et al. [21], case-based training could be beneficial for nurses in the prevention of delirium.

The introduction of preventive measures and early detection is crucial for controlling delirium, but also for avoiding associated adverse events such as falls [33], therapeutic errors and pressure ulcers, which increase the average length of stay and affect the functionality of these people [34]. 

Although the aim of this SR is not to explore professionals’ decision making, diagnosis, and instruments used for the early detection of delirium, the results showed that this syndrome is often misdiagnosed [1,5,13,35].

Timely diagnosis and management may curtail the duration and associated morbidity and mortality of delirium [36] or even reverse symptoms and give the critically ill individual a better quality of life, avoiding prolonged hospitalizations and/or other comorbidities [34,35]. 

The assessment of delirium needs to be carried out at the bedside by a multidisciplinary team. Several studies have shown that both intensivists and ICU nurses perform poorly when using clinical judgement alone, resulting in many missed cases of delirium [35,36]. There are various scoring systems developed assess the risk of delirium, the most commonly used and validated tools are the Confusion Assessment Method for the Intensive Care Unit (CAM-ICU) and the Intensive Care Delirium Screening Checklist (ICDSC) [35]. To this end, the multidisciplinary team, and nurses in particular, must recognize that the ideal clinical practice is one that ensures evaluation of the risk, early detection and harmonious coordination between pharmacotherapy and non-pharmacological strategies.

The limitations of this SR are related to the methodological option of restrictions placed on the language and free access to full text, excluding some articles that may have respected the inclusion criteria. It should also be noted that the studies are heterogeneous and it was decided not to assess their methodological quality. The fact of not having performed the assessment of the methodological quality of studies and the risk of bias limits recommendations for the clinic. It is suggested that effectiveness studies be carried out for the identified interventions, especially non-pharmacological ones, to increase the evidence and strength of recommendations for the clinic.

## 5. Conclusions

This literature review discussed non-pharmacological preventive and management interventions that can be integrated into the daily practice of caring for people in critical situations. Several subcategories were identified, including the promotion of a routine sleep–wake cycle; environmental control; the daily and constant reorientation of the individual; the involvement of the family in care; cognitive and memory stimulation; sensory stimulation; early mobilization; the promotion of comfort; management of pain and anxiety; and prevention of other complications. The results found are in line with what is recommended in the ABCDEF bundle, a reference in preventive care for the development of delirium in ICU patients. In addition to the interventions mentioned, it was possible to see that training nurses is equally important, so that the risk can be correctly evaluated, and the necessary preventive measures implemented. 

Despite the lack of scientific evidence proving its efficacy, pharmacological therapy should be considered in the prevention and management of delirium. Important clinical practice guidelines, in particular the PADIS guidelines and the ABCDEF bundle, corroborate this assumption. After analyzing the results obtained, we conclude that psychoactive drugs should be used carefully in critically ill individuals, at the lowest dose and for the shortest time possible. In addition to the benefits, pharmacological administration must take into account the main adverse effects. 

## Figures and Tables

**Figure 1 healthcare-12-01134-f001:**
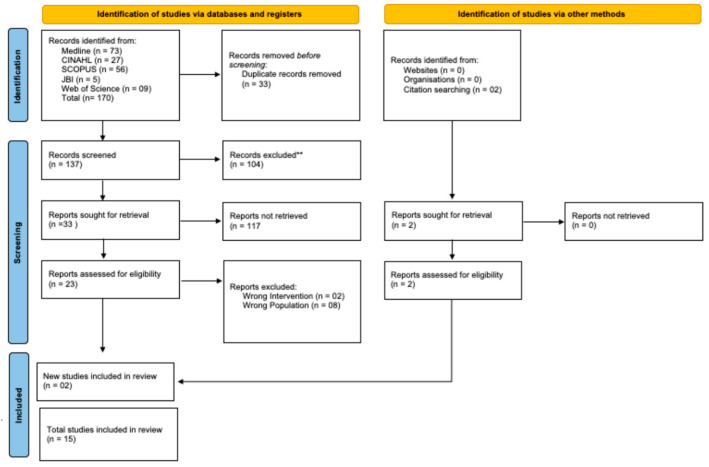
PRISMA 2020 flowchart. Lisbon, 2023.

**Table 1 healthcare-12-01134-t001:** Eligibility criteria. Lisbon, 2023.

	Inclusion Criteria	Exclusion Criteria
P	Adults (aged 18 or over) in critical condition.	Adults hospitalized in services other than intensive care
C	Interventions for the prevention of delirium; Interventions for the management of delirium; Pharmacological and non-pharmacological interventions.	Studies to validate scales for assessing delirium; Epidemiological studies on delirium; Risk factors for delirium.
C	Intensive care unit.	Emergency, nursing ward, or community.

**Table 2 healthcare-12-01134-t002:** Search strategy used in the Medline database search. Lisbon, 2023.

	Search Strategy	N
#1	((((((((((((((((((((((adult [Title/Abstract]) OR (young adult [Title/Abstract])) OR (older adult[Title/Abstract])) OR (older person [Title/Abstract])) OR (elder* [Title/Abstract])) OR (eldery [Title/Abstract])) OR (age* [Title/Abstract])) OR (aged [Title/Abstract]))) OR (grown up[ Title/Abstract])) OR (adult [MeSH Terms])) OR (aged [MeSH Terms])) OR (adult, frail older [MeSH Terms])) OR (adults [MeSH Terms])) OR (adults, young [MeSH Terms])) OR (adult, young [MeSH Terms])) OR (elder, frail [MeSH Terms])) NOT (neonate [MeSH Terms])) NOT (children [MeSH Terms]) NOT (adolescent [MeSH Terms])) NOT (animals [MeSH Terms])) NOT (paediatric [Title/Abstract])) NOT (pediatric [Title/Abstract])) AND ((ffrft[Filter]) AND (English[Filter] OR Portuguese [Filter] OR Spanish [Filter]) AND (2018:2023 [pdat]))	241,452
#2	(“delirium” [Title/Abstract] OR “acute confusion” [Title/Abstract] OR “acute delirium” [Title/Abstract] OR “delirium management” [Title/Abstract] OR “delirium prevention” [Title/Abstract] OR “delirium” [MeSH Terms] AND ((ffrft[Filter]) AND (English[Filter] OR Portuguese [Filter] OR Spanish [Filter]) AND (2018:2023 [pdat])))	4714
#3	((((((((critically ill [Title/Abstract]) OR (critical illness [Title/Abstract])) OR (critical patient [Title/Abstract])) OR (critical care patient [Title/Abstract])) OR (critically ill [MeSH Terms])) OR (critical illness [MeSH Terms])) OR (critical care [MeSH Terms])) OR (care, critical [MeSH Terms])) OR (critical illnesses [MeSH Terms]) AND ((ffrft[Filter]) AND (English[Filter] OR Portuguese [Filter] OR Spanish [Filter]) AND (2018:2023 [pdat]))	21,994
#4	((((((“prevent*” [Title/Abstract] OR “prevention” [Title/Abstract] OR “preventing” [Title/Abstract] OR “treat*” [Title/Abstract] OR “treating” [Title/Abstract] OR “treatment” [Title/Abstract] OR “manage” [Title/Abstract] OR “management” [Title/Abstract] OR “managing” [Title/Abstract] OR “manage*” [Title/Abstract] OR “nursing intervention” [Title/Abstract] OR “intervention*” [Title/Abstract] OR “nursing care” [Title/Abstract] OR “care” [Title/Abstract] OR “critical care nursing” [Title/Abstract] OR “pharmacological interventions” [Title/Abstract] OR “non pharmacological interventions” [Title/Abstract] OR “autonomous nursing interventions” [Title/Abstract] OR “interventions” [Title/Abstract] OR “patient care management” [MeSH Terms]) OR “nursing care” [MeSH Terms]) OR “pharmacologic actions” [MeSH Terms]) OR “critical care” [MeSH Terms]) OR “control [Title/Abstract] NOT “nursing assessment” [MeSH Terms] NOT “risk assessment” [MeSH Terms]) NOT “risk factors” [MeSH Terms] NOT “causality” [MeSH Terms]) NOT (prevalence [MeSH Terms])) NOT (incidence [MeSH Terms]) AND (ffrft[Filter]) AND (English[Filter] OR Portuguese [Filter] OR Spanish[Filter]) AND (2018:2023[pdat])	1,922,251
#5	(((“intensive care unit” [Title/Abstract] OR “intensive care” (Title/Abstract] OR “intensive care units” (MeSH Terms]) NOT “intensive care units, neonatal” [MeSH Terms]) NOT “intensive care units, pediatric” [MeSH Terms]) NOT “intensive care, neonatal” (MeSH Terms]) NOT “nurseries, infant” (MeSH Terms]) NOT “ambulatory care” [MeSH Terms]) NOT “emergency medical services” [MeSH Terms] AND ((ffrft[Filter]) AND (English[Filter] OR Portuguese [Filter] OR Spanish [Filter]) AND (2018:2023 [pdat])))	43,329
#6	#1 AND #2 AND #3 AND #4 AND #5	73

**Table 3 healthcare-12-01134-t003:** Final bibliographic sample. Lisbon, 2023.

Author (Year) Country	Study Type	Objective	Conclusions
[2] (2021) USA	Narrative literature review	Describe the current state of scientific evidence for diagnostic, preventive, and therapeutic measures capable of mitigating the course of delirium.	Antipsychotics, dexmedetomidine, statins, and ketamine are not recommended to prevent delirium. Antipsychotics can be considered in the short-term treatment of delirium in the context of controlling severe agitation or other stressful symptoms.
[13] (2018) USA	Randomized, double-blind, placebo-controlled	Determine the effect of haloperidol or ziprasidone in the treatment of delirium during a diagnosis of critical illness.	The use of haloperidol or ziprasidone, compared to a placebo, did not significantly alter the duration of delirium.
[3] (2020) USA	Book chapter	Present the clinical characteristics, assessment, prophylactic strategies, and treatment of delirium in the ICU context.	Haloperidol is especially useful in the treatment of hyperactive delirium. Dexmedetomidine is useful in mechanically ventilated adults when hyperactive delirium makes weaning difficult. Benzodiazepines are only useful in individuals with a history of alcoholism or for delirium resulting from seizures. Early mobilization can reduce the duration and incidence of delirium.
[14] (2020) South Korea	Randomized, double-blind, placebo-controlled	Evaluate the efficacy of low-dose, short-term quetiapine in preventing delirium in critically ill patients.	The prophylactic use of low-dose quetiapine can be useful in preventing delirium in critically ill patients.
[15] (2019) Germany	Multinational, prospective, cohort	Evaluate the prevalence and variables associated with the use of haloperidol in the treatment of delirium in the ICU, and explore any association between the use of haloperidol and 90-day mortality.	Haloperidol was the most common pharmacological intervention for the treatment of delirium, regardless of its subtype. Benzodiazepines, quetiapine, dexmedetomidine, and olanzapine were used to treat delirious patients admitted to the ICU.
[16] (2021) Iran	Randomized, double-blind, placebo-controlled	Evaluate the efficacy and safety of valerian and lemongrass, in addition to quetiapine, in critically ill patients diagnosed with delirium and agitation.	The addition of Neurogol (a combination of valerian and lemongrass) to quetiapine can reduce agitation and shorten the length of ICU stay, without adverse effects.
[17] (2022) USA	Restrospective, cohort	Evaluate the association between selective serotonin reuptake inhibitors (SSRIs) and delirium within 24 h of pharmacological administration in critically ill adults.	The administration of SSRIs is associated with a decreased risk of delirium within 24 h of administration in adult patients admitted to a medical or surgical ICU.
[18] (2022) USA	Literature review	Check whether the scientific evidence supports the hypothesis that non-pharmacological interventions are more effective in preventing and managing delirium than drug therapy.	With less reliance on pharmacological therapy and greater awareness of non-pharmacological interventions, we can try to use less therapy and more personal care, promoting the dignity of patients and minimizing delirium.
[5] (2021) Portugal	Scoping review	Identify possible nursing interventions that can be implemented in the prevention and management of delirium in adults in the ICU.	It was possible to identify different categories of nursing interventions for the prevention and management of delirium in adults and elderly people admitted to the ICU, namely pharmacological and non-pharmacological..
[19] (2020) Canada	Study Protocol	Determine the effect of prevention, detection, and management of delirium by the family of critically ill individuals on symptoms such as anxiety and depression, compared to usual care.	Family involvement in the prevention, detection, and management of delirium has the potential to produce positive effects. Family members can observe subtle changes in behavior/cognition and can be involved in non-pharmacological measures, such as reorientation and mobilization activities.
[20] (2022) Poland	Perspective article	Discuss the current burden of delirium in the ICU, as well as recommendations and forecasts in the management of admitted patients, environmental changes, and infrastructure adaptations that will contribute to a delirium-free ICU.	The presence of advanced neuromonitoring will allow for better control of anxiety, pain, agitation, sleep, and prevention of delirium. Suggestions for a healthy environment include separation of ICU equipment, including alarms and monitors, from the person, to allow for noise control; presence of natural light; contact with nature; presence of television and other devices that allow the patient’s cognitive training; connection with family and friends through video calls; presence of guiding elements (such as clocks and calendars, family photos); and early mobilization. Education about delirium in the ICU, including screening and eliminating potential modifiable risk factors.
[21] (2022) China	Qualitative study	Explore critical care nurses’ perceptions of current non-pharmacological practices in the prevention of delirium in adult ICUs, including delirium screening, early mobilization, sleep promotion, family involvement, and sensory stimulation.	Case-based training could be beneficial for nurses in preventing delirium. The adoption of a sensory stimulation protocol was another strategy highlighted by nurses to reduce the incidence and severity of delirium.
[22] (2019) USA	Pragmatic randomized clinical trial	Test the effectiveness of applying a pharmacological management of delirium (PMD) bundle in improving and reducing the severity of delirium in ICU patients.	The PMD bundle was not effective in reducing the duration and severity of delirium.
[23] (2019) South Korea	Review article	Discuss the overview of delirium and update the literature with reference to the 2018 PADIS guidelines.	Examples of non-pharmacological strategies for the prevention and management of delirium include regular anxiety relief; patient orientation; reducing environmental noise and the use of alarms; using light consistent with the day–night circadian cycle; and promoting mobility. The application of the ABCDEF bundle is associated with a reduction in delirium. Haloperidol may be considered in the treatment of acute agitation (hyperactive delirium). The use of dexmedetomidine may be a better strategy for the treatment of delirium than the administration of benzodiazepines. The prophylactic administration of dexmedetomidine in the evening results in a delirium prevention rate of 80%.
[24] (2021) India	Prospective, randomized, parallel-group clinical trial	Investigate the effectiveness of a new delirium prevention bundle for mechanically ventilated critically ill people.	The implementation of a new delirium prevention bundle in the ICU does not significantly reduce the incidence of delirium when compared to the standard intervention protocol for mechanically ventilated patients.

## Data Availability

Data are available only upon request to the authors.

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
