# Peer review of "Nursing Intervention to Prevent and Manage Delirium in Critically Ill Patients: A Scoping Review"

_healthcare, 2024, doi:10.3390/healthcare12111134_

Round 1
Reviewer 1 Report
Comments and Suggestions for Authors
This is a review article on the pharmacologic and non-pharmacologic strategies to prevent and treat delirium. The article is focused on nursing education and intervention. In this article they did a literature review of open access journals and found 170 articles, 33 articles were read, and 15 articles met inclusion criteria. The article then discussed the main points in these articles first non-pharmacological interventions and then pharmacological interventions. The discussion then synthesized this information.
One limit of this review article is the limited number of articles included. It would be helpful to explain the reasons behind the exclusion criteria. The article did have a table on exclusion criteria, but it would be helpful to know why they chose this exclusion criteria. It did seem like a significant number of articles were excluded and may need an additional comment.
The article did make a general summary of the articles but it would be helpful to have a more in depth analysis to include more data/conclusions from the articles reviewed. I have seen this in a table form in other studies. Examples:
Line 158-164 discussed sleep habits and the authors could elaborate on what the studies showed on outcomes improved by sleep (what percentage of patients benefited/what was the reported reduction in delirium).
It would also be helpful to sharing the data in amount of reduction of delirium for all other discussed non-pharmacological interventions (noise, reorientation of patients, memory stimulation, sensory stimulation, early mobilization, etc).
Line 202-206 I felt the comment about the lack of training for nurses was off topic and not sure why it is included here and maybe better in the discussion section.
Line 207-216 The ABCDF bundle was discussed in multiple articles reviewed and maybe should have more elaboration on the data of the outcomes reported. How much improvement in survival, length of stay, duration of delirium, etc).
Line 272 “Haloperidol is the drug of choice for its (delirium) treatment” and Line 275 “Haloperidol as a first-line pharmacological agent in the treatment of delirium is no longer suggested…” are contradictory and confusing. Maybe a better explanation is needed such as: “Haloperidol is the drug of choice but despite that it is no longer recommended as the first line agent..” but I am not sure if this is the point the authors are trying to make.
Line 290 mentioned hypoactive delirium and no further mention of treatment options for it and how it is treated differently.
318-325 mention other medications used to treat delirium in this part of the discussion section but these medications were not mentioned in the studies evaluated. Line 319-321 mentions multiple drugs that can be given to prophylactically prevent delirium but only one of these medications was mentioned in section 3.2.1 Pharmacological prevention of delirium in the ICU (line 240)
The article did not focus on the diagnosis of delirium but alludes to the importance of diagnosis and therefore early treatment in the discussion section. I think it is important at least to mention why diagnosis can be difficult or delayed since it is an important part of the process.
Explain line 363 (maybe a language barrier) “Studies are needed on the skills of professionals for the prevention and management of delirium” What skills? Maybe education? Barriers to implementing these interventions?
Line 360 The reviewers did not assess the methodological quality of the studies. What did they review? I think a very big limitation and should be elaborated on why and how this can affect your conclusions
TABLES:
Table 2 (search strategy) is confusing #6 says using search strategy 1,2,3,4,5 lead to 73 articles and the result section mentioned 137 analyzed and 104 excluded. Maybe there need to be a better discussion of the search process for the articles other then just the table.
Line 89 and 102: what is “Lisbon, 2023”?
Comments on the Quality of English Languageminor edits/clarifications needed see details above
Author Response
We attached the answer.

Reviewer 2 Report
Comments and Suggestions for Authors
Dear author, thank you for your work
I would like to make a series of observations to improve its quality.
a) the exclusion criteria should be reformulated; for example, being a pediatric population cannot be considered an exclusion criterion as it is not eligible "per se", since the inclusion criterion itself does not allow it.
b) identify the researchers who performed the full text reading of the selected articles.
c) in case of discrepancy among the investigators regarding the value of an article selected, were there any criteria to establish the tie-breaker? were the investigators in agreement in all decisions?
d) were criteria established for analyzing the quality of the scientific papers, as in Caspe? why was this not done?
Could you implement this requirement?
Thank you
Author Response
We attached the answer.
